# Identification of Epigenetic Interactions between MicroRNA-30c-5p and DNA Methyltransferases in Neuropathic Pain

**DOI:** 10.3390/ijms232213994

**Published:** 2022-11-13

**Authors:** Raquel Francés, Jorge Mata-Garrido, Roberto de la Fuente, María Carcelén, Miguel Lafarga, María Teresa Berciano, Raquel García, María A. Hurlé, Mónica Tramullas

**Affiliations:** 1Departamento de Fisiología y Farmacología, Facultad de Medicina, Universidad de Cantabria, 39011 Santander, Spain; 2Instituto de Investigación Sanitaria Valdecilla (IDIVAL), 39011 Santander, Spain; 3Departamento de Anatomía y Biología Celular, Universidad de Cantabria, 39011 Santander, Spain; 4Centro de Investigación Biomédica en Red Sobre Enfermedades Neurodegenerativas (CIBERNED), 28029 Madrid, Spain; 5Servicio de Anestesiología, Hospital Universitario Marqués de Valdecilla, 39008 Santander, Spain; 6Departamento de Biología Molecular, Universidad de Cantabria, 39011 Santander, Spain

**Keywords:** neuropathic pain, miR-30c-5p, epigenetic, TGF-β1, DNA methylation, DNMTs

## Abstract

Neuropathic pain is a prevalent and severe chronic syndrome, often refractory to treatment, whose development and maintenance may involve epigenetic mechanisms. We previously demonstrated a causal relationship between miR-30c-5p upregulation in nociception-related neural structures and neuropathic pain in rats subjected to sciatic nerve injury. Furthermore, a short course of an miR-30c-5p inhibitor administered into the cisterna magna exerts long-lasting antiallodynic effects via a TGF-β1-mediated mechanism. Herein, we show that miR-30c-5p inhibition leads to global DNA hyper-methylation of neurons in the lumbar dorsal root ganglia and spinal dorsal horn in rats subjected to sciatic nerve injury. Specifically, the inhibition of miR-30-5p significantly increased the expression of the novo DNA methyltransferases DNMT3a and DNMT3b in those structures. Furthermore, we identified the mechanism and found that miR-30c-5p targets the mRNAs of DNMT3a and DNMT3b. Quantitative methylation analysis revealed that the promoter region of the antiallodynic cytokine TGF-β1 was hypomethylated in the spinal dorsal horn of nerve-injured rats treated with the miR-30c-5p inhibitor, while the promoter of Nfyc, the host gene of miR-30c-5p, was hypermethylated. These results are consistent with long-term protection against neuropathic pain development after nerve injury. Altogether, our results highlight the key role of miR-30c-5p in the epigenetic mechanisms’ underlying neuropathic pain and provide the basis for miR-30c-5p as a therapeutic target.

## 1. Introduction

Acute pain is a defence mechanism from damaging processes, which usually disappears after healing. However, in susceptible individuals, pain can persist for months or even years, even though the original injury may have long since disappeared. Chronic pain is a severe burden on patients’ daily lives and constitutes a significant public health issue worldwide with a significant socio-economic impact [1]. The factors conditioning predisposition or resilience to transition from acute to chronic pain constitute an important gap in our knowledge [2].

Neuropathic pain is a prevalent (7–10% of the general population) and debilitating chronic pain syndrome caused by injuries or diseases, affecting the somatosensory nervous system. Major causes of neuropathic pain include trauma, metabolic disorders (diabetes), infections (herpes zoster, HIV), neurotoxic agents (chemotherapy), and idiopathic (trigeminal neuralgia) [3]. In vulnerable individuals, neural damage, even if transient, can promote activity-dependent, long-lasting maladaptive remodelling of neurons, glial and inflammatory cells, synapses, and circuits at many different levels of the nervous system. This maladaptive remodelling results in hyperexcitability, spontaneous activation of the nociceptive pathway, and dysfunction of endogenous pain-modulation systems [1]. The typical manifestations of neuropathic pain are exaggerated responses to painful stimuli (hyperalgesia) and dysesthetic painful sensations (tingling, burning, pricking, electric shock), either spontaneously or triggered by innocuous stimuli (allodynia), which can persist for years [4]. Seventeen percent of individuals suffering from pain with neuropathic characteristics have a health-related quality of life score equivalent to “worse than death” (International Association for the Study of Pain, IASP). Unfortunately, the management of neuropathic pain is an unmet clinical need as the analgesic efficacy of currently available drugs is very limited. Moreover, we are far from effective, mechanism-based, disease-modifying therapies [5,6].

Compelling evidence supports the idea that epigenetic events, including DNA methylation, post-translational modifications of histones, and expression changes in non-coding RNAs, are critical contributors to the transition from acute to chronic pain in preclinical models and clinical settings [7,8,9,10]. The reprogramming of the DNA methylation status by neurons and glial cells in response to neural injuries may explain the persistent transcriptional changes within the spinal dorsal horn (SDH), dorsal root ganglia (DRG), and other nociception-related brain areas, which result in chronic neuropathic pain [11].

The epigenetic post-transcriptional regulation of specific pain-related genes by microRNAs (miRNAs) is emerging as a novel therapeutic avenue for managing chronic pain conditions [12,13]. Our previous experimental and clinical studies revealed a pivotal etiopathogenic role for miR-30c-5p in neuropathic pain development and maintenance [14]. Thus, the spared injury of the sciatic nerve (SNI) in rats induces an overexpression of miR-30c-5p in nociception-related areas of the nervous system, including the SDH and DRG, and circulating in cerebrospinal fluid (CSF) and plasma, which correlates directly with the severity of the allodynia developed by the rats after SNI. Importantly, the early treatment of rats with a cycle of an miR-30c inhibitor (100 ng/10 µL) into the cisterna magna at the time of SNI and on days 4 and 7 after surgery delays the development of neuropathic pain and prevents miR-30c-5p upregulation in the nervous system (SDH), as well as in CSF and plasma. The uptake of the miR-30c-5p inhibitor by spinal cells is evidenced by injecting a fluorescent-labelled miRNA inhibitor (200 ng/10 mL) into the cisterna magna. In addition, the administration of an miR-30c-5p mimic accelerates and exacerbates neuropathic pain. The clinical relevance of these findings was highlighted in a cohort of patients with critical leg ischemia, in which elevated levels of miR-30c-5p in plasma and CSF discriminated, with high sensitivity and specificity, patients with neuropathic pain symptoms from patients without neuropathic pain [14].

Recent studies reported coordinated actions between miRNAs and other epigenetic mechanisms to reinforce the regulation of gene expression [15,16]. In particular, miRNAs can modify DNA methylation patterns by directly targeting DNA methyltransferases (DNMTs) [17,18,19]. Meanwhile, the aberrant methylation of the promoter regions of miRNAs can influence miRNA transcription with a subsequent impact on their mRNA targets [20,21].

We hypothesise a cross-talk between the modulation exerted by miR-30c-5p, at both the post-transcriptional and transcriptional level, and DNA methylation patterns within the somatosensory nervous system may result in neuropathic pain development. A better understanding of epigenetic plasticity might provide new prospects to prevent pain chronification after injury and promote neuropathic pain recovery. The present study aimed to elucidate whether miR-30c-5p overexpression after nerve injury modifies DNA methylation patterns in DRGs and SDH, resulting in neuropathic pain.

## 2. Results

### 2.1. The miR-30c-5p Inhibitor Prevents Neuropathic Pain after Sciatic Nerve Injury in Rats

To validate the nocifensive consequences of pharmacological modulation of miR-30c-5p, a series of rats subjected to SNI or sham surgery received an early intracisternal cycle of either an miR-30c-5p-inhibitor or a random anti-miR sequence (100 ng/10 µL on days 0, 4, and 7 after SNI). The nocifensive responses to mechanical stimuli were determined every two days with von Frey monofilaments (Appendix A). SNI rats treated with random anti-miR (*n* = 5) exhibited mechanical allodynia with maximal severity on day 10 after SNI. In contrast, at the same time point after SNI, the rats treated with miR-30c-5p-inhibitor (*n* = 5) were fully protected against neuropathic pain (*p* < 0.001).

### 2.2. Changes in Global DNA Methylation Induced by Sciatic Nerve Injury within Spinal Dorsal Horn and Dorsal Root Ganglia Neurons and Its Modification by Treatment with miR-30c-5p-Inhibitor

We carried out an immunofluorescence study using an antibody against 5′-MeC in dissociates of lumbar SDH and DRG neurons obtained from rats treated with random anti-miR or miR-30c-5p-inhibitor, 10 days after either SNI or sham surgery (*n* = 20 neurons per animal, and *n* = 3 rats per group).

As shown in Figure 1 and Appendix A, the nuclei of SDH and DRG neurons from sham rats treated with either miR-30c-5p-inhibitor or random anti-miR exhibited similarly low 5′-MeC immunoreactive signals (Figure 1A,B,E,F). Importantly, both the 5′-MeC mean signal intensity (SDH: sham + random anti-miR (0.99 ± 0.3, *n* = 2) vs. 10 days-SNI + random anti-miR (3.42 ± 0.18, *n* = 3) *p* < 0.001; 10 days-SNI + miR-30c-5p-inhibitor (16.61 ± 1.02, *n* = 3) vs. 10 days-SNI + random anti-miR (3.42 ± 0.18, *n* = 3) *p* < 0.001, and DRG: sham + random anti-miR (0.99 ± 0.19, *n* = 3) vs. 10 days-SNI + random anti-miR (2.93 ± 0.27, *n* = 3) *p* < 0.001; 10 days-SNI + miR-30c-5p-inhibitor (11.69 ± 0.59, *n* = 3) vs. 10 days-SNI + random anti-miR (2.93 ± 0.27, *n* = 3) *p* < 0.001) and the percentage of the nuclear area immunolabeled (SDH: sham + random anti-miR (23.26 ± 6.64, *n* = 2) vs. 10 days-SNI + random anti-miR (48.43 ± 3.5, *n* = 3) *p* < 0.01; 10 days-SNI + miR-30c-5p-inhibitor (87.34 ± 4.2, *n* = 3) vs. 10 days-SNI + random anti-miR (48.43 ± 3.5, *n* = 3) *p* < 0.01, and DRG: sham + random anti-miR (14.66 ± 1.43, *n* = 3) vs. 10 days-SNI + random anti-miR (41.19 ± 5.03, *n* = 3) *p* < 0.001; 10 days-SNI + miR-30c-5p-inhibitor (76.45 ± 3.07, *n* = 3) vs. 10 days-SNI + random anti-miR (41.19 ± 5.03, *n* = 3) *p* < 0.001) increased significantly after SNI and were even stronger in SNI rats treated with miR-30c-5p-inhibitor (Figure 1C,D,G–L).

These results indicated that SNI was accompanied by a global DNA hyper-methylation status in the first- (DRG) and second-order (SDH) neurons of the nociceptive pathway. Furthermore, treatment with miR-30c-5p-inhibitor selectively induced a strong global DNA hyper-methylation in DRG and SDH neurons from rats subjected to SNI, but not in sham rats. In the former case, 5′-MeC immunofluorescences showed a nuclear non-homogeneous distribution pattern (Figure 1C,D,G,H). It included areas of diffuse signal (Figure 1C,D,H), the brightest heterochromatin domains around the nucleolus (“perinucleolar chromatin”) and in the nuclear interior (Figure 1C), and the negative staining of the nucleolus and, presumably, of chromatin-free nuclear compartments, such as nuclear speckles and Cajal bodies (Figure 1C,G,H) [22].

### 2.3. Changes in the Expression of DNA Methyltransferases Induced by Sciatic Nerve Injury within Spinal Dorsal Horn and Dorsal Root Ganglia Neurons and Its Modification by miR-30c-5p-Inhibitor Treatment

DNMTs are the enzymes responsible for DNA methylation at the C-5 position of cytosine residues. In mammalian cells, the maintenance DNMT1 enzyme methylates newly synthesised DNA during replication, and the de novo DNMT3a and DNMT3b catalyse the addition of new methylation marks at CpG sites [23,24,25].

Our next objective was to assess if the changes in global DNA methylation observed in SDH and DRG neurons from SNI rats were associated with parallel changes in the expression (mRNA and protein) of de novo DNMTs.

As shown in Figure 2, the treatment of sham rats with miR-30c-5p-inhibitor did not change the expression of any de novo DNMT in the SDH or DRG. SNI rats treated with random anti-miR, compared with sham rats, featured a significant upregulation of DNMT3b (sham + random anti-miR (2.48 ± 0.5, *n* = 5) vs. 10 days-SNI + random anti-miR (4.19 ± 0.45, *n* = 5), *p* < 0.05) in the SDH (Figure 2B) and of DNMT3a (“10 days-SNI + miR-30c-5p-inhibitor (224.056 ± 60.84, *n* = 4) vs. 10 days-SNI + random anti-miR (1344.83 ± 320.17, *n* = 4), *p* < 0.01) and DNMT3b (sham + random anti-miR (3.86 ± 1.89, *n* = 4) vs. 10 days-SNI + random anti-miR (8.89 ± 2.23, *n* = 7), *p* < 0.05) in the DRG (Figure 2E,F).

SNI rats treated with miR-30c-5p-inhibitor exhibited significantly higher gene expression levels of both DNMTs than SNI rats treated with a random anti-miR, in both the SDH (DNMT3b (10 days-SNI + miR-30c-5p-inhibitor (9.46 ± 2.4, *n* = 5) vs. 10 days-SNI + random anti-miR (4.19 ± 0.45, *n* = 5), *p* < 0.05)) (Figure 2A,B) and DRG (DNMT3a (10 days-SNI + miR-30c-5p-inhibitor (224.056 ± 60.84, *n* = 4) vs. 10 days-SNI + random anti-miR (1344.83 ± 320.17, *n* = 4, *p* < 0.01); DNMT3b (10 days-SNI + miR-30c-5p-inhibitor (22.83 ± 4.34, *n* = 4) vs. 10 days-SNI + random anti-miR (8.89 ± 2.23, *n* = 7), *p* < 0.01)) (Figure 2E,F). As expected, SNI rats treated with miR-30c-5p-inhibitor exhibited significantly higher protein expression levels in the SDH of DNMT3a (sham + random anti-miR: 1.001 ± 0.001; SNI + random anti-miR: 0.94 ± 0.004; SNI + miR-30c-5p inhibitor: 1.20 ± 0.066, *p* < 0.05, *n* = 2) and DNMT3b (sham + random anti-miR: 1.005 ± 0.005; SNI + random anti-miR: 1.125 ± 0.005; SNI + miR-30c-5p inhibitor: 1.83 ± 0.069, *p* < 0.001, *n* = 2) (Figure 2C,D) than SNI rats treated with a random anti-miR and in the DRG ((DNMT3a: sham + random anti-miR: 1.00 ± 0.0004; SNI+ random anti-miR: 1.337 ± 0.312; SNI + miR-30c-5p inhibitor: 8.14 ± 0.709, *p* < 0.05, *n* = 2); (DNMT3b: sham + random anti-miR: 1.84 ± 0.145; SNI + random anti-miR: 5.40± 3.14; SNI + miR-30c-5p inhibitor: 25.58 ± 0.53, *p* < 0.01, *n* = 2)). Together, these results indicated that DNMT regulation was congruent with the global DNA methylation status of the neurons of each treatment group.

### 2.4. Luciferase Reporter Assays Showed Post-Transcriptional Regulation of De Novo DNMT3a and DNMT3b by miR-30c-5p in HeLa Cells

The existence of mutual feedback modulatory loops between specific miRNAs, including miR-30c-5p, and DNMTs has been documented [17,26]. Thus, we assessed whether de novo DNMTs are post-transcriptionally regulated by miR-30c-5p, as suggested by our neurochemical data. To this end, DNMT3a and DNMT3b pLightSwitchTM or pGL3-REPORT™ luciferase miRNA expression reporter vectors, containing the predicted binding sequence for miR-30c-5p in their respective 3′UTR region, were transiently transfected into human HeLa human cell lines. Cells were co-transfected with an miR-30c-5p mimic or a scramble miR mimic in parallel experiments (two independent assays with quintupled measurements). Luciferase activity was assessed 24 h thereafter. The luciferase reporter assays further confirmed that DNMT3a and DNMT3b were validated mRNA targets of miR-30c-5p in HeLa cells. As shown in Figure 3, the miR-30c-5p mimic significantly decreased the luciferase activity after cell transfection with 25 ng of either DNMT3a-3’UTR (3′UTR + scramble miR mimic (1.14 ± 0.12, *n* = 2); 3′UTR +miR-30c-5p mimic (0.56 ± 0.09, *n* = 2), *p* < 0.05) (Figure 3A) or DNMT3b-3′UTR (3′UTR + scramble miR mimic (0.88 ± 0.13, *n* = 2); 3′UTR +miR-30c-5p mimic (0.0075 ± 0.0005, *n* = 2), *p* < 0.001) (Figure 3B), compared with cells transfected with the scramble miR-mimic (DNMT3a-3’UTR (pGL3 + scramble miR mimic (1.0 ± 0.0, *n* = 2); pGL3 + miR-30c-5p mimic (1.23 ± 0.03, *n* = 2)) or DNMT3b-3′UTR (pLight + scramble miR mimic (1.0 ± 0.0, *n* = 2); pLight +miR-30c-5p mimic (1.26 ± 0.0007, *n* = 2)).

### 2.5. Modulation of miR-30c-5p in Rats Subjected to Sciatic Nerve Injury Results in Methylation Changes of Its Host Gene Nfyc

The miR-30c-1 is a mirtron embedded within the Nuclear Transcription Factor Y Subunit Gamma (Nfyc)-coding host gene. Mirtrons and their host genes are frequently transcribed under the control of the same promoter [27,28]. Therefore, we evaluated a coordinated transcriptional regulation of the mirtron miR-30c-1 and its host gene Nfyc. Linear regression and correlation analyses indicated that Nfyc mRNA levels in the SDH (Figure 4A) and DRG (Figure 4D) correlated directly with miR-30c-5p. Moreover, Nfyc and miR-30c-5p expressions featured parallel changes under the different experimental conditions of our study in both the SDH (Figure 4B,C) (Nfyc: sham + random (118.5 ± 7.05, *n* = 4); 10 days-SNI + random (223.7 ± 33.57, *n* = 5); 10 days-SNI + miR-30c-5p-inhibitor (126.6 ± 16.25, *n* = 4); 5 days-SNI + miR-30c-mimic (186 ± 15.85, *n* = 5), *p* < 0.05, miR-30c-5p: sham + random (9.83 ± 1.5, *n* = 5); 10 days-SNI + random (23.87 ± 5.0, *n* = 4); 10 days-SNI + miR-30c-5p-inhibitor (2.64 ± 0.65, *n* = 5); 5 days-SNI + miR-30c-mimic (2.46 ± 0.65, *n* = 5), *p* < 0.001) and DRG (Figure 4E,F) (Nfyc: sham + random (13.13 ± 4.75, *n* = 5); 10 days-SNI + random (54.88 ± 13.41, *n* = 5) *p* < 0.01; 10 days-SNI + miR-30c-5p-inhibitor (12.33 ± 2.45, *n* = 5); 5 days-SNI + miR-30c-mimic (63.37 ± 5.28, *n* = 5), *p* < 0.001; miR-30c-5p: sham + random (2.17 ± 0.96, *n* = 4); 10 days-SNI + random (8.13 ± 2.06, *n* = 4); 10 days-SNI + miR-30c-5p-inhibitor (1.97 ± 0.74, *n* = 5); 5 days-SNI + miR-30c-mimic (6.94 ± 0.89, *n* = 5), *p* < 0.01).

This parallel transcription of Nfyc and miR-30c-5p in the SDH and DRG under SNI conditions led us to assess whether silencing miR-30c-5p could modulate DNA methylation in the promoter region of the gene-encoding Nfyc using the methylation-sensitive qPCR technique (ms-qPCR) [29]. Quantitative analysis (Figure 5A) revealed that, compared with random anti-miR, the treatment of the SNI rats with the miR-30c-5p inhibitor produced a strong increase in the percentage of methylation of the Nfyc promoter in the SDH (sham + random anti-miR (3.34 ± 1.0, *n* = 5); 10 days-SNI + random anti-miR (3.8 ± 1.0, *n* = 5); 10 days-SNI + miR-30c-5p-inhibitor (8.29 ± 1.47, *n* = 4, *p* < 0.05)), which supports epigenetic autoregulation of miR-30c-5p transcription. Thus, the hyper-methylation of the Nfyc promoter would result in transcriptional repression and, consequently, in the negative regulation of both miR-30c-5p and Nfyc, as observed in SNI rats treated with the miR-30c-5p inhibitor (Figure 4B,C). However, the amount of sample needed for genomic DNA extraction precluded performing these experiments in DRGs.

### 2.6. Modulation of miR-30c-5p in Rats Subjected to Sciatic Nerve Injury Results in Methylation Changes of Its Target TGF-β1

Previous results from our group showed that the anti-inflammatory cytokine TGF-β1 constitutes a direct post-transcriptional target of miR-30c-5p, which mediates the antiallodynic effect of the miR-30c-5p inhibitor in SNI rats [14]. Therefore, we considered it to be of great interest to explore the impact of miR-30c-5p modulation on the epigenetic control of TGF-β1 transcription in SNI rats. As shown in Figure 5B, the percentage of methylation of the TGF-β1 promoter increased significantly in neuropathic SNI rats treated with a random anti-miR (sham + random anti-miR (0.36 ± 0.21, *n* = 4); 10 days-SNI + random anti-miR (2.93 ± 0.87, *n* = 3); 10 days-SNI + miR-30c-5p-inhibitor (0.077 ± 0.057, *n* = 4), *p* < 0.01), in comparison with sham rats. In contrast, pain-free SNI rats treated with the miR-30c-5p inhibitor showed promoter hypo-methylation, compared to neuropathic SNI rats. Therefore, our results supported that, in addition to its canonical post-transcriptional effects, miR-30c-5p can indirectly regulate the methylation status of the TGF-β1 promoter.

## 3. Discussion

Neuropathic pain is one of the most complex human diseases caused by a lesion or dysfunction of the nervous system. A growing body of evidence reveals the contribution of complex epigenetic mechanisms to persistent gene expression alterations within the somatosensory nervous system, which are responsible for transitioning from acute to chronic neuropathic pain after nerve injury [30,31,32]. MiRNAs are post-transcriptional regulators of gene expression under physiological and pathological conditions, and their deregulation in chronic pain syndromes has been widely explored [14,33,34,35]. In addition, de novo DNMT3a and DNMT3b play a major role in DNA methylation at previously unmethylated CpG islands [36,37]. Recent evidence supported the involvement of DNMT3a deregulation in chronic pain conditions [38,39]. The particular roles of individual epigenetic mechanisms in neuropathic pain have been widely explored; however, the cross-talk between miRNAs and other epigenetic marks that govern neuropathic pain have barely been investigated. The study of such interactions may depict a more complex layer of gene regulation [40] under neuropathic pain conditions; for example, (i) the expression of miRNAs could be regulated by multiple epigenetic mechanisms [41], (ii) miRNAs could repress the expression of epigenetic factors [42], and (iii) miRNAs and epigenetic factors could cooperate to modulate common targets [43].

In the present study, we demonstrated an interplay between miR-30c-5p expression and chromatin modifications via the regulation of DNMT3a and DNMT3b, which may contribute to the persistent aberrant signalling in nociception-related pathways underlying neuropathic pain. 

First, our 5′-MeC immunofluorescence results revealed a robust increase in the methylated form of cytosine in the DNA of nociception-related neurons from SNI animals. The 5′-MeC immunofluorescence constitutes a good approach to estimating global DNA methylation that, in eukaryotes, mainly occurs at CpG dinucleotides and is frequently associated with transcriptional repression [43]. We demonstrated that 5′-meC distributed throughout the nucleus, excluding the nucleolus and irregular areas that presumably correspond to chromatin-free nuclear compartments, such as nuclear speckles of splicing factors and Cajal bodies [44]. Accumulations of 5′-MeC, particularly in the perinucleolar region, a nuclear domain enriched in genes important for cellular stress response and sensory perception [45], were detected in DRG and SDH neurons. Interestingly, pain-free SNI rats (treated with the miR-30c-5p inhibitor) exhibited significantly higher global DNA methylation than neuropathic SNI rats treated with vehicle.

DNA hyper-methylation after SNI was consistently associated with overexpression of de novo DNMT3a and DNMT3b in DRG and SDH neurons. These results suggested that the increased DNA methylation observed in DRG and SDH neurons after SNI may be mediated via DNMT3b and, to a lesser extent, DNMT3a. Furthermore, in SNI rats treated with the miR-30c-5p inhibitor, the DNA hyper-methylation observed in isolated DRG and SDH neurons with 5′-meC was paralleled by a marked upregulation of DNMT3a and DNMT3b in lysates from both tissues, which could be related to new methylation patterns induced by the anti-miR in neurons and, probably, in non-neuronal cells.

Our findings were consistent with previously published data showing DNMT3a and DNMT3b overexpression in neurons of the DRG and/or SDH using several models of sciatic nerve injury in mice and rats [39,46,47,48,49,50], chemotherapy-induced neuropathic pain [51], and inflammatory pain models [52]. However, in contrast to our results, [53] did not find differences in DNMT3b expression in the DRG of rats subjected to L5 spinal nerve ligation (SNL). This inconsistency could be accounted for by differences in the experimental model used as the SNI used in our study is more harmful than SNL.

Overall, these results agreed with our initial hypothesis that epigenetic modifications of chromatin can stabilise aberrant gene expression believed to trigger/perpetuate neuropathic pain. Since DNA methylation is reversible and can be modulated by chemical agents, our results further highlight the de novo DNA methylation as an essential epigenetic mark for the development of novel pain relief medications, and as an unexplored mechanism involved in the analgesic effect of old drugs (i.e., opioids, NSAIDs, antidepressants, etc.) [54]. Restoring DNA methylation via the inhibition of DNMTs may represent an effective strategy to reverse nerve injury-induced neuropathic pain by recovering the expression of pain suppressor genes that were silenced by methylation. Additionally, an early intervention to avoid epigenetic changes after acute nerve injury might prevent the progression to a chronic pain state. Interestingly, we showed that inhibiting miR-30c-5p induces a hyper-methylation state and increases, in parallel, DNMT3a and DNMT3b expression levels in the DRG and SDH after SNI. Therefore, we suggest that the pharmacologically induced knockdown of miR-30c-5p results in analgesia after SNI by a mechanism involving DNA hyper-methylation via the de novo DNMTs and subsequent transcriptional repression of genes promoting neuropathic pain.

In contrast, the concurrent upregulation of miR-30c-5p [14] and de novo DNMTs in untreated SNI rats did not fit into the canonical relationship between an miRNA and its target mRNA. Indeed, in complex pathophysiological processes, such as neuropathic pain, the alteration of gene expression involves a much more intricate interaction between players. Thus, under the neuropathic pain condition, numerous miRNAs, besides miR-30c-5p, could be contributing to regulating the expression of DNMTs. In turn, DNMTs could control the transcription of hundreds of miRNAs.

The relationship between neuropathic pain development and methylation changes in specific genes induced by nerve injury remains mostly unknown. In this regard, it was reported that DNMT3a overexpression in the DRG and/or SDH may contribute to neuropathic pain development after sciatic nerve injury by promoting DNA hyper-methylation in the promoter of genes encoding the opioid receptors mu and kappa (Oprm1, Oprk1) and voltage-gated potassium channels (Kcna2) [40,49,50,54,55]. On the other hand, Jiang et al. [56,57] reported a downregulation of DNMT3b in the spinal cord after SNL, which results in demethylation of the promoter regions of the chemokine receptor CXCR3 and the G-protein-coupled receptor 151 (GPR151). Furthermore, in a recent study, Liu et al., 2020 [43] reported that DNMT3a expression was upregulated and associated with the hyper-methylation of the miR-214-3p promoter in the dorsal horn in a rat spinal nerve ligation model. As a result, the expression of CSF1, an miR-214-3p target gene, was enhanced in astrocytes, which led to the neuroinflammation and mechanical allodynia observed after spinal nerve injury.

We further analysed whether transcripts encoding de novo DNMTs constituted direct targets of miR-30c-5p using the luciferase reporter assay, a very reliable method for determining the capacity of an miRNA to decrease luciferase activity in cells when it binds to its mRNA target [55]. We checked the interaction between miR-30c-5p and DNMT3b, predicted by online software, and DNMT3a, as suggested by our in vivo results and others [56]. Our results showed an apparent reduction in luciferase activity when cells were co-transfected with the miR-30c-5p mimic and plasmids containing the 3’UTR regions of either DNMT3a or DNMT3b. Consequently, our in vitro results indicated that both DNMT3a and DNMT3b were direct targets of miR-30c-5p, confirming the prediction of online software regarding DNMT3b and adding DNMT3a to the list of validated miR-30c-5p targets, as suggested by the experimental data (present results and Ref. [56]).

Herein, we assessed whether DNMT3a and DNMT3b regulation by miR-30c-5p might contribute to a differential DNA methylation profile in the promoter regions of pain-related genes, explaining the long-lasting antiallodynic effects observed after the inhibition of miR-30c-5p.

In this regard, our first object of study was the host gene of miR-30c-5p, which encodes Nfyc. In humans, nearly half of the known miRNAs are encoded within the introns of protein-coding genes. However, it is unclear whether mirtrons are transcriptionally linked to their host genes or transcribed independently [55]. Herein, we observed that Nfyc mRNA and miR-30c-5p expressions presented parallel changes under the different experimental conditions of our study with a positive linear relationship. We also showed that treatment with the miR-30c-5p inhibitor reduced the expression values of Nfyc in SNI rats, while the miR-30c-5p mimic tended to increase them. These results suggested that miR-30c-5p can either directly or indirectly regulate the expression of its host gene in the SDH and DRG.

If Nfyc were directly targeted by miR-30c-5p, a feature not predicted by the online informatics algorithms, the post-transcriptional regulation would have taken place in the opposite direction to that observed here (i.e., miR-30c-5p upregulation would induce Nfyc downregulation). Given that our results supported DNMT3a and DNMT3b as miR-30c-5p targets, we hypothesised an indirect transcriptional regulation of Nfyc via de novo DNA methylation. Supporting this idea, the methylation-sensitive qPCR analysis indicated that the CpG sites in the promoter region of Nfyc were hypermethylated in SNI rats treated with the miR-30c-5p inhibitor. Under miR-30c-5p inhibition, de novo DNMT3s would recover their expression, allowing them to methylate the promoter region of Nfyc. The resultant transcriptional repression of Nfyc and, subsequently, miR-30c-5p in nociception-related areas may contribute to the long-lasting antiallodynic effect produced by treatment with the miR-30c-5p inhibitor in SNI rats (Appendix A).

Several examples in the literature report autoregulatory mechanisms of mirtrons on the expression and functions of their host genes, either via positive or negative feedback loops [58,59,60]. However, it should be borne in mind that the cross-talk between miR-30c-5p and its host gene could occur via many other mechanisms, including other miRNAs, depending on tissues and conditions.

The TGF-β is a family of pleiotropic, contextually acting cytokines [61], which plays an essential role in nociceptive processing [62]. In particular, evidence obtained from animal models supports that TGF-β1 prevents the neuronal plasticity underlying pain sensitisation and promotes the activation of endogenous pain inhibitory pathways [14,62,63,64,65]. Our previous studies prove that TGF-β1 is a target of miR-30c-5p in the SDH of rodents under neuropathic pain conditions. Moreover, TGF-β1 downregulation after SNI is a key mechanism underlying sensitisation to nociceptive stimuli, whereas TGF-β1 gain-of-function results in allodynia relief or prevention [14]. Therefore, we further analysed the regulatory networks between miR-30c-5p and TGF-β1 through epigenetic mechanisms.

Our results presented herein showed that the SDH from rats subjected to SNI featured aberrant CpG island DNA hyper-methylation in the promoter region of the gene-encoding TGF-β1, associated with neuropathic pain. In contrast, pain-free SNI rats treated with miR-30c-5p inhibitor showed promoter hypo-methylation. This evidence suggests that the methylation state of the TGF-β1 promoter may be regulated by miR-30c-related mechanisms, with long-term consequences on the pain-related behaviours of SNI rats.

Altogether, our data suggested that the long-lasting antiallodynic effect induced by miR-30c-5p inhibitor was associated with strong global DNA hyper-methylation in neurons of the dorsal root ganglia and spinal dorsal horn, accompanied by upregulation of DNA methyltransferases. Thus, these epigenetic marks resulted in hyper-methylation of the Nfyc promoter and, consequently, transcriptional repression of Nfyc, the host gene of miR-30c-5p, in the spinal dorsal horn, while the promoter of the antiallodynic cytokine TGF-β1 appeared hypo-methylated, resulting in long-term protection against the development of neuropathic pain. Our results highlight the key role of miR-30c-5p in the epigenetic plasticity underlying neuropathic pain through the existence of a cross-talk between miR-30c -5p and DNA methylation within the somatosensory nervous system and provide new insights into preventing pain chronification after injury and promoting neuropathic pain recovery. Further studies are warranted to identify differentially methylated genes under neuropathic pain conditions. Thus, DNA methylome studies of DRG in animal models of neuropathic pain and of human peripheral blood mononuclear cells (PBMC) in patients with neuropathic pain and pain-free controls will allow us to unravel mechanisms potentially involved in the development of the disease and to identify biomarkers with potential value for the diagnostics and/or prognostics of neuropathic pain.

## 4. Materials and Methods

### 4.1. Animals

The experiments were performed in 8-week-old (250 to 300 g) male Sprague Dawley rats. All animals were kept under controlled conditions of humidity and temperature (22 ± 1 °C, 60–70% relative humidity) on a 12 h light/dark cycle (lights on at 8:00 a.m.). Food and water were supplied ad libitum. Under anaesthesia with isoflurane (Forane^®^, 1.5–2.5%, 30% N20, and 70% O_2_), animals were sacrificed by decapitation or perfused with a freshly prepared solution containing 3.7% paraformaldehyde (Merck, 158127, Madrid, SpainA) in HEPEM buffer (2xHPEM: HEPES 60 mM, Pipes 130 mM, EGTA 20 mM, and MgCl2.6H2O 4 mM, pH 6.9) containing 0.5% Triton X-100 through the left ventricle using a 30G needle, according to the experimental procedure. The study was approved by the University of Cantabria Institutional Laboratory Animal Care and Use Committee (reference IP0415) and conducted following the guidelines of directive 2010/63/ EU of the European Parliament and the International Association for the Study of Pain.

### 4.2. Study Design

Here, we assessed the role and influence of miR-30c-5p in the aberrant DNA methylation that follows peripheral nerve injury (Appendix A). We performed studies in rats subjected to SNI and in cultured cells. The following experimental studies were designed: (i) rats subjected to SNI and injected with a miR-30c-5p inhibitor or a random anti-miR to identify the influence of miR-30c-5p inhibition on global DNA methylation levels and the relative expression of the two main DNMTs (DNMT3a and DNMT3b) in the SDH and DRGs via immunofluorescence studies, qPCR, and Western blot. Via qPCR, we assessed (ii) a coordinated transcriptional regulation of miR-30c-a and its host gene Nfyc, and (iii) the percentage methylation of two pain-related genes (TGF-β1 and Nfyc). In addition, the existence of feedback modulatory loops between miR-30c-5p and DNMT3a, and DNMT3b was assessed by the luciferase assay using HeLa cells.

In each experimental series, the surgical intervention and administration of the treatments were performed by an operator who was not involved in the subsequent behavioural performance. Each animal was coded and the researcher who carried out the behavioural experiment was not aware of their group allocation. The investigator who analysed the nocifensive behavioural data did know which animals were grouped together but was blinded as to the treatment these groups received.

### 4.3. Neuropathic Pain Model

Chronic neuropathic pain was induced using the spared nerve injury (SNI) model [66]. Briefly, rats were anaesthetised with isoflurane (Forane^®^, 1.5–2.5%, 30% N20, and 70% O_2_), and the left common sciatic nerve was exposed at the level of its trifurcation. The tibial and common peroneal branches of the nerve were sectioned, and the sural nerve was left untouched. Sham-operated rats underwent the same procedure, but all sciatic branches were left intact. The development of neuropathic pain was assessed using the manual von Frey test. After a period of adaptation of 10 min to their new habitat, the plantar surface of the paw was stimulated with a series of von Frey monofilaments (Semmes Weinstein von Frey Aesthesiometer for Touch Assessment, Stoelting Co., Wood Dale, IL USA) graduated on a scale based on the force, expressed as mN (millinewtons), which they produce. The following monofilaments were applied: 5.9 mN, 9.8 mN, 13.7 mN, 19.6 mN, 39.2 mN, 58.8 mN, 78.4 mN, 98.0 mN, 147 mN, 255 mN, 588 mN, and 980 mN. Shaking, licking, or withdrawal of the paw after applying the mechanical stimulus was considered a positive response. The paw surface of the animal was stimulated six times with each of the monofilaments and the percentage of positive responses was evaluated. Initially, the paw was stimulated with a monofilament of intermediate strength, and monofilaments of progressively decreasing force were used until no response was obtained. Subsequently, stimuli of ascending force were applied, starting from the monofilament consecutive to the initial, until 100% responses were provoked [67]. 

### 4.4. Treatment and Experimental Groups

Rats were anaesthetised with isoflurane (Forane^®^, 1.5–2.5%, 30% N20, and 70% O_2_) and placed on a stereotaxic frame for intracisternal administration of miR-30c-5p inhibitor (100 ng/10µL, mirVana MH11060, Thermo Fisher Sci., Waltham, MA, USA) diluted in a mixture of Lipofectamine and artificial CSF. A series of rats received a cycle of three intracisternal injections of miR-30c-5p inhibitor at the time of the SNI or sham intervention and on days four and seven after surgery. Rats were sacrificed when maximal differences in allodynia between groups were observed (at day 10 after nerve injury). Sham animals received intracisternal injections composed of artificial CSF, random anti-miR, and lipofectamine following an identical protocol. Animals were sacrificed by decapitation under deep anaesthesia with 2% isoflurane. The spinal cord was quickly extracted by hydro-extrusion; the lumbar SDH and DRGs (L3, L4, and L5) were dissected. Samples were frozen in dry ice and stored at −80 °C until further use.

### 4.5. Immunofluorescence and Confocal Microscopy: 5′-MeC Detection

Rats were deeply anaesthetised with 2,2,2-Tribromoethanol (Aldrich T48402, St. Louis, MO, USA) administered intraperitoneally at a dose of 25 mg/kg and perfused with a freshly prepared solution containing 3.7% paraformaldehyde (Merck 158127) in HEPEM buffer containing 0.5% Triton X-100. After perfusion, the lumbar SDH and DRGs were collected and washed three times in HEPEM buffer. Squash preparations of dissociated neurons were performed as previously described [68]. The samples were incubated overnight with a mouse anti-5′-methylcytosine primary antibody (1:100, Eurogentec 091105, Seraing, Belgium) at 4 °C. Subsequently, the slides were washed with 0.05% Tween 20 in PBS and incubated for 45 min at room temperature with the specific FITC-conjugated secondary antibody (339038 Jackson ImmunoResearch Europe LTD, Ely, UK). Finally, the sections were rinsed in PBS and mounted using Vectashield (Vector Laboratories, Newark, CA, USA, H-1000-10). Confocal imaging was performed with an LSM510 (Zeiss, Aalen, Germany) laser scanning microscope and using a 63× oil (1.4 NA) objective. Images were obtained by excitation at 488 mm to detect FITC and processed using Adobe Photoshop CS4 software (Adobe Systems Inc., v 20.0.0, Mountain View, CA, USA). At least 20 neurons per animal were examined (*n* = 3 animals per experimental condition). The quantitative analysis of immunofluorescence signal intensities was performed using ImageJ software (NIH, Bethesda, MD, USA; http://rsb.info.nih.gov/ij/, accessed on 15 March 2020). Images were acquired using the same confocal settings, with fluorescent signals at the brightest cells being non-saturated. The values were corrected for background staining by subtraction of a blank measurement taken outside the cell. Data were represented by mean fluorescence intensity and relative to control samples. Images were processed using Adobe Photoshop CS4 software (Adobe Systems Inc., v 20.0, Mountain View, CA, USA).

### 4.6. RNA Extraction and Real-Time RT-PCR

RNA was isolated with Trizol, following the manufacturer’s instructions (Invitrogen, Carlsbad, CA, USA, 15596026). One µg of RNA was reverse-transcribed to cDNA using a High-Capacity cDNA Reverse Transcription Kit (Life Technologies, Carlsbad, CA, USA, 4368814) and random primers. The expression levels of the mRNA candidates, DNMT3a, DNMT3b, Nfyc, and TGF-β1, were determined by qPCR using gene-specific SYBRGreen (Takara, Shiga, Japan, RR820A)-based primers (0.4 µM): DNMT3a: forward 5′-CCGGGTGCTATCTCTCTCTTTG-3′ and reverse 5′-TGACGATGGAGAGGTCATTG-3′; DNMT3b: forward 5′-GCAAGAGAGAGGCCCTCAG-3′ and reverse 5′-TGTGAGGGAGATGCTCAGTG-3′; Nfyc: forward 5′-CCTGTATCAGGCACCCAAGT-3′ and reverse 5′-GGTGACTTGCTGGATCTGGT-3′; 18S: forward 5′-ACCGCAGCTAGGAATAAGGA-3′ and reverse 5′-GCCTCAGTTCCGAAAACCA-3′; Nfyc (ms-qPCR): forward 5′-TGACCAATAAGGTGCCAGGT-3′ and reverse 5′-CGCCATGTTGTGTCTTCG-3′; TGF-β1 (ms-qPCR): forward 5′-GATCCTCCAGACAGCTAGGC-3′ and reverse 5′-ACTCCTCCTCCCCCTCCT-3′. The mRNA expression was normalised using 18S. All samples were run in triplicate and each qPCR was performed in duplicate. Expression was portrayed as mean ± SEM.

### 4.7. Western Blot

Immunoblotting was carried out in whole-cell SDH lysates of SDH and DRG. Equal amounts of whole-cell protein (50 µg) were resolved on a 12.5% SDS-polyacrylamide gel and transferred to a polyvinylidene difluoride membrane using standard procedures. The following primary antibodies were used: rabbit anti-DNMT3a (Abcam, ab 2851) 1:500, rabbit anti-DNMT3b (Abcam, sc 374015) 1:1000, and rabbit anti Lamb1 (Abcam, ab 108536) 1:1000, followed by the secondary antibody at a 1:10,000 dilution (IRDye680DX, Rockland Immunochemicals 926-68071). Immunoreactivity was detected with an OdysseyTM Infrared-Imaging System (LI-COR Biosciences, Lincoln, NE, USA), according to the OdysseyTM Western Blotting Protocol. The integrated optical density of the bands was determined using Image J software (U.S. National Institutes of Health, Bethesda, MD, USA, http://imagej.nih.gov/ij, accessed on 1 June 2020). The nuclear marker laminin-1β (Lamb1) was used as an internal loading control.

### 4.8. Methylation Sensitive qPCR (ms-qPCR)

Genomic DNA from the lumbar SDH was extracted using NucleoSpin^®^ Tissue (Macherey–Nagel, Düren, Germany, 740952.50). Approximately 25 mg of tissue were lysed by incubation with proteinase K/SDS solution for 3 h at 56 °C. The samples were loaded onto the NucleoSpin^®^ Tissue Column, and DNA was collected after several washes following the manufacturer’s instructions. An amount of 300 ng of genomic DNA was digested with CpG-methylation-sensitive enzymes: TauI (3 U/μL; ER1651 ThermoFisher Scientific) or FauI (200 U/µL; R0651S New England BioLabs, Ipswich, MA, USA); or with non-CpG-methylation-sensitive enzymes: SacI (10 U/μL; ER1132 ThermoFischer Scientific) or HphI (10 U/µL; ER1102 ThermoFischer Scientific) for two hours with 600 rpm shaking, at the manufacturer’s suggested temperature. SYBR Green-based qPCR was carried out in triplicate with a total volume of 20 μL per tube containing 1 μL of genomic DNA (TauI-digested, SacI-digested, or undigested DNA), 0.4  μL of each primer, 10  μL of SYBR Select Master Mix (Life Technologies 4472913), and 8.2 μL of H_2_O. The percentage of methylation was determined from the change in Ct. The relationship of ΔCt to percentage methylation was calculated using the formula %me = 100 × 10^−0.7(ΔCt)^. The following primers were used: Nfyc: forward 5′-TGACCAATAAGGTGCCAGGT-3′ and reverse 5′-CGCCATGTTGTGTCTTCG-3′; TGF-β1: forward 5′-GATCCTCCAGACAGCTAGGC-3′ and reverse 5′-ACTCCTCCTCCCCCTCCT-3′.

### 4.9. Cell Culture and Luciferase Assay

HeLa cells (00194CP, Cell Line Service) were cultured at a density of 10,000 cells per well in 96-well dishes with Dulbecco’s Modified Eagle Medium F12 (DMEM/F12, Gibco, 11320033) supplemented with 10% foetal bovine serum (Biological Industries, Haemek, Israel A4766801), 100 U/mL penicillin (Gibco, Carlsbad, CA, USA, 15140148), and 100 U/mL streptomycin (Gibco 15140148). Cells were cultured for 24 h at 37 °C in a humified atmosphere containing 5% CO_2_. After 24 h, the growth medium was removed and replaced with Optimem (Gibco, Life Technologies 31985062). Cells were transfected using the X-tremeGENE 9 DNA transfection reagent (Sigma-Aldrich 6365787001) with pGL3-3′-UTR-DNMT3a (3′UTR DNMT3a fragment of 1500 bp donated by Dr. Letizia Pitto’s laboratory [56], 25 nM) or pLightSwitch-3′UTR-DNMT3b (3′UTR DNMT3b fragment of 1525 bp, 25 nM, SwitchGear Genomics, Carlsbad, CA, USA, S809202). Empty pGL3 or pLightSwitch plasmids were used as a control. Twenty-four hours later, cells were transfected with miR-30c-5p mimic (10 nM) (mirVana MC11060, Thermo Fisher Scientific) or scramble miR mimic (4464058, Thermo Fisher Scientific), according to the manufacturer’s protocol. After 24 h, cells were washed twice with PBS and the luciferase assay was performed using the Dual Luciferase Reporter Assay System. Firefly luciferase activity was normalised to the total amount of protein for each transfected well. Two independent experiments were performed (five wells per condition).

### 4.10. Statistical Analysis

GraphPad Prism 5.01 (Domatics, San Diego, USA), Predictive Analytics SoftWare (PASW) 22 (Statistical Package for the Social Sciences (SPSS) Inc.) (IBM, New York, USA), and Stata 14/SE (StataCorp, College Station, TX, USA) packages were used. The values were expressed as mean ± SEM. Data from the behavioural study was analysed by mixed-design three-way ANOVA (Split-Plot ANOVA) followed by the Bonferroni post hoc test, as there were three factors involved; force intensity (within-subject factor), treatment (between-subject factor) and nerve injury (between-subject factor) and the first factor (force intensity) was a repeated measure. Differences between multiple groups in 5′-MeC mean signal intensity and percentage of the methylated area were analysed by two-way ANOVA followed by the Bonferroni post hoc test, in which there were two factors, treatment and nerve injury. For the comparisons of gene/protein expression or Luc activity, where values were between more than two groups, one or two-way ANOVA followed by the Bonferroni post hoc test was used. Correlations between mRNA and miRNA expression values were performed using Pearson’s correlation analysis. A *p*-value of less than 0.05 was considered statistically significant.

## Figures and Tables

**Figure 1 ijms-23-13994-f001:**
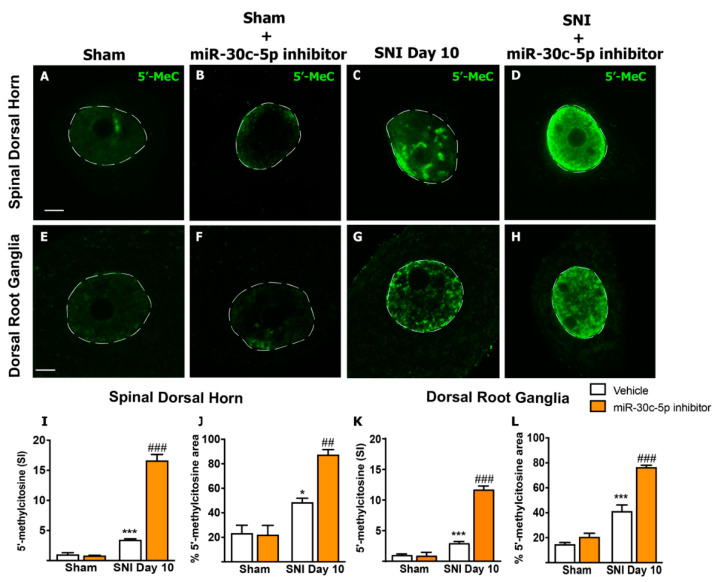
Spinal dorsal horn and dorsal root ganglia neurons from rats treated with either random anti-miR or miR-30c-5p-inhibitor exhibit global DNA hyper-methylation after sciatic nerve injury. Representative images showing 5′-methylcytosine (5′-MeC) positive immunostaining in neurons isolated from the spinal dorsal horn (SDH) (**A**–**D**) and dorsal root ganglia (DRG) (**E**–**H**) from sham rats treated with random anti-miR (**A**,**E**), sham rats treated with miR-30c-5p-inhibitor (**B**,**F**), 10 days-SNI rats treated with random anti-miR (**C**,**G**), and 10 days-SNI rats treated with miR-30c-5p-inhibitor (**D**,**H**). The 5′-MeC mean signal intensity (SI) (**I**,**K**) and the percentage of the methylated area (**J**,**L**) were determined in 60 neurons (20 neurons per rat, 3 rats per group). Values are mean ± SEM. * *p* < 0.05, *** *p* < 0.001 vs. sham; ## *p* < 0.01, ### *p* < 0.001 vs. SNI (two-way ANOVA followed by Bonferroni post hoc test). Scale bar: 5 µm.

**Figure 2 ijms-23-13994-f002:**
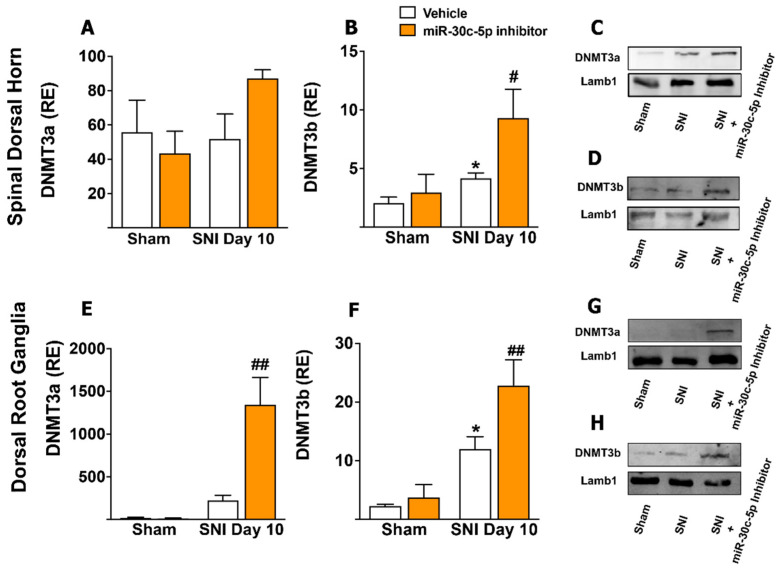
Effects of the treatment with miR-30c-5p inhibitor on the expression of de novo DNA methyltransferases in the dorsal root ganglia and spinal dorsal horns from rats subjected to sciatic nerve injury. The mRNA relative expression (RE) of DNMT3a and DNMT3b determined by qPCR and normalised to 18S in the lumbar spinal dorsal horn (SDH) (**A**,**B**) and dorsal root ganglia (DRG) (**E**,**F**) from sham rats treated with random anti-miR (*n* = 6), sham rats treated with miR-30c-5p inhibitor (*n* = 5), SNI rats treated with random anti-miR (*n* = 8), and SNI rats treated with miR-30c-inhibitor (*n* = 6). Values are mean ± SEM. * *p* < 0.05 vs. sham; # *p* < 0.05, ## *p* < 0.01, vs. SNI (two-way ANOVA followed by Bonferroni post hoc test). Protein expression levels of DNMT3a, DNMT3b, and the nuclear marker laminin-1β (Lamb1) determined by Western blot in SDH (**C**,**D**) and DRG (**G**,**H**) lysates from sham rats (*n* = 2), SNI rats (*n* = 2), and SNI rats treated with miR-30c-inhibitor (*n* = 2). * *p* < 0.05, vs. sham; # *p* < 0.05, ## *p* < 0.01 vs. SNI (one-way ANOVA followed by Bonferroni post hoc test).

**Figure 3 ijms-23-13994-f003:**
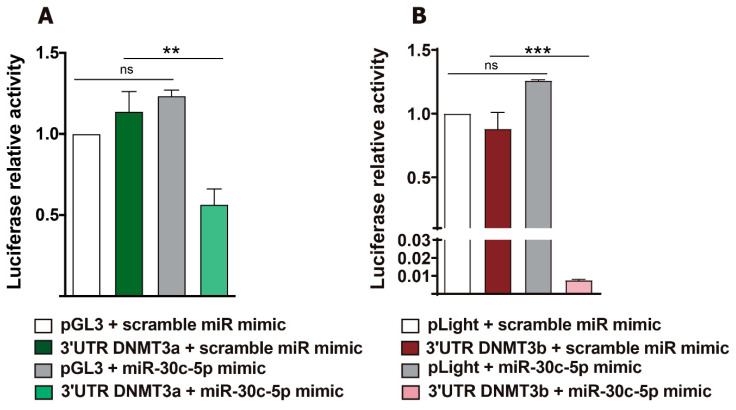
Post-transcriptional regulation of de novo DNMTs by miR-30c-5p in cultured cells. Luciferase reporter assays in HeLa cells co-transfected with pGL3-REPORT luciferase vector (25 ng) containing the 3′UTR of DNMT3a (**A**) and pLight-REPORT luciferase vector containing the 3′UTR of DNMT3b (**B**) and miR-30c-5p mimic (10 nM). The data represent the mean ± SEM of the relative luciferase units normalised to the amount of protein in two independent experiments with quintupled measurements.; n.s, not significant, ** *p* < 0.01, *** *p* < 0.00 vs. 3’UTR + scramble miR mimic (one-way ANOVA followed by Bonferroni post hoc test).

**Figure 4 ijms-23-13994-f004:**
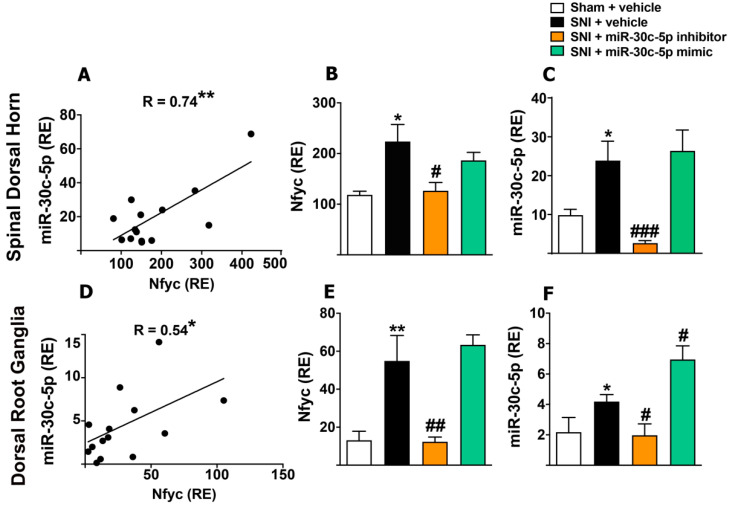
Effects of the treatment with miR-30c-5p inhibitor or miR-30c-5p mimic on the expression of miR-30c-5p and its host gene Nfyc in the spinal dorsal horn and dorsal root ganglia from rats subjected to SNI. (**A**,**D**): Linear regression and Pearson′s correlation analyses show the relationship of miR-30c-5p relative expression with Nfyc mRNA levels in the spinal dorsal horn (SDH) and dorsal root ganglia (DRG). r: Pearson′s correlation coefficient. Nfyc mRNA (**B**,**E**) and miR-30c (**C**,**F**) expression in sham and SNI rats treated with random, miR-30c-5p mimic, or miR-30c-5p inhibitor. (*n* = 5 rats per group.) Values are mean ± SEM. * *p* < 0.05, ** *p* < 0.01 vs. sham; # *p* < 0.05, ## *p* < 0.01, ### *p* < 0.001 vs. SNI (one-way ANOVA followed by Bonferroni post hoc test).

**Figure 5 ijms-23-13994-f005:**
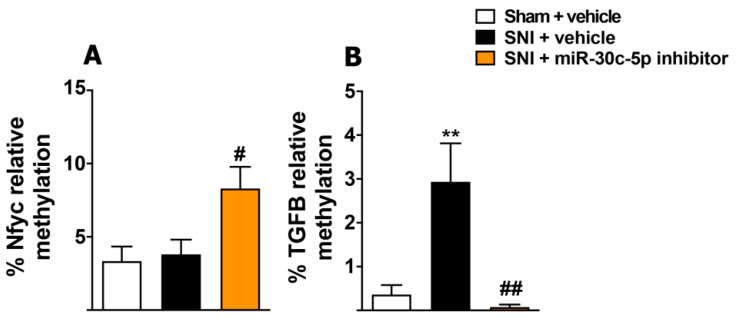
Effects of the treatment with miR-30c-5p inhibitor or miR-30c-5p mimic on the methylation of the promoters of Nfyc and TGF-β1 in the spinal dorsal horn from rats subjected to SNI. Percentage of methylation of the promoters of genes coding Nfyc (**A**) and TGF-β1 (**B**) in the spinal dorsal horn (SDH) from SNI rats treated with random anti-miR or miR-30c-5p inhibitor. (*n* = 5 rats per group). Values are mean ± SEM. ** *p* < 0.01 vs. sham; # *p* < 0.05, ## *p* < 0.01 vs. SNI. One-way ANOVA followed by Bonferroni post hoc test.

## Data Availability

The data presented in this study are available in the article.

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
