# Peer review of "Identification of Epigenetic Interactions between MicroRNA-30c-5p and DNA Methyltransferases in Neuropathic Pain"

_ijms, 2022, doi:10.3390/ijms232213994_

Round 1

Reviewer 1 Report

The work by Frances et al., provides an advancement for the understanding of the mechanism of neuropathic pain in a SNI rat model. The paper is well structured and fairly well written. However, some aspects that could improve the quality have been detected as follows starting by major considerations:

The data is presented with some numbers which are not clear to the reader if they correspond to means or medians of the n=xx/group measures (for example, “treatment”: F(1,7) = 117.7, p<0.001; “nerve injury”: F(1,7) = 231.8, p<0.001), and SD values are missing. Supplementary information with raw values is encouraged.

Section 2.4 seems to need extra controls to show the activity of DNMT3 3´UTR Luc constructs with scrambled miR-30c-5p. The presence of the extra DNMT3 sequence could turn into decreased Luc activity by mechanisms independent of miR-30c-5p and therefore the conclusions reached by the researchers in this experiment are considered inaccurate.

It seems important to specify miR or inhibitor dose applied, and the route used. How did the authors ensure delivery and levels of miR-30c-5p-inhibitor to SDH and RDG? Please, define exactly the inhibitor used, how was applied and monitored in destiny (target tissue) for an improved understanding of the work performed.

Methods need more detail. Most products miss to indicate catalogue numbers.

Minor considerations:

Although Fig. 1 is probably shown representative illustration, additional images should be available as supplementary Figures.

It is not clear if the authors subcloned DNMTs 3´UTRs in commercial vectors or whether commercial products were available as used. The authors need to explain the reasons to support the use of 2 different Luciferase constructs. Also, the positions of the 3´UTRs of the referred genes in their corresponding mRNAs (was the whole sequence or a fragment of it?) and the restriction sites used in the subcloning.

On line 203 the sentence “DNMT3a and DNMT3b were direct mRNA targes of miR-30c-5p in HeLa cells.” has to be rephrased as there is no evidence for a direct mechanism when the variable measured is luciferase activity.

Explain RE and ROD abbreviations in Fig. 2

Please indicate the n for each group in Fig. 2 legend, either 2 or 3 in each case (a range does not seem appropriate).

Intronic miRNAs are generally referred as mirtrons.

Fig. 4 misses to indicate the n for each reading.

Abbreviations should be used consistently after their introduction. Point 2.3 can be shortened by applying this rule.

Line 155 It is not “the maintenance that methylates” it is the enzyme.

Latin terms such as “de novo” should be italicized. Please review the whole sentence on lines 154 and following.

Please check typo in “Merk”.

Author Response

The work by Frances et al., provides an advancement for the understanding of the mechanism of neuropathic pain in a SNI rat model. The paper is well structured and fairly well written. However, some aspects that could improve the quality have been detected as follows starting by major considerations:

Many thanks for your valuable comments to help us to improve our manuscript.

Major considerations

Point 1: The data is presented with some numbers which are not clear to the reader if they correspond to means or medians of the n=xx/group measures (for example, “treatment”: F(1,7) = 117.7, p<0.001; “nerve injury”: F(1,7) = 231.8, p<0.001), and SD values are missing. Supplementary information with raw values is encouraged.

Response 1: We apologize for omitting such important information in the text. Following the reviewer's indications, we have included the mean, standard error of the mean (SEM) and the N values in all the results shown in the manuscript. Furthermore, all F values have been now included in a“ Statistical analysis” in Supplementary material. In addition, an excel file with the raw data from the study has been included in the Supplementary material.

Point 2: Section 2.4 seems to need extra controls to show the activity of DNMT3 3´UTR Luc constructs with scrambled miR-30c-5p. The presence of the extra DNMT3 sequence could turn into decreased Luc activity by mechanisms independent of miR-30c-5p and therefore the conclusions reached by the researchers in this experiment are considered inaccurate.

Response 2: We have addressed the reviewer`s query in the new version of the manuscript by adding new controls to show the Luc activity of DNMT3 3´UTR in the presence of a scramble miR-mimic. The results section (pages 6 to 7, lines 239 to 243) and the new Figure 3 (page 7, line 245) have been modified accordingly by including the new experimental conditions.

Point 3: It seems important to specify miR or inhibitor dose applied, and the route used. How did the authors ensure delivery and levels of miR-30c-5p-inhibitor to SDH and RDG? Please, define exactly the inhibitor used, how was applied and monitored in destiny (target tissue) for an improved understanding of the work performed.

Response 3: Following the reviewer’s indication, we have added a more detailed description of the miR-30c-5p inhibitor used in the study (dose, route of administration, catalogue number) as well as an explanation related to the delivery and levels reached in destiny (Introduction section: page 2, lines 79 to 85).

Please note that, our group has focused during the last years on the study of the role of miRNA-related mechanisms triggered by peripheral nerve damage, which contribute to neuropathic pain development.

Point 4: Methods need more detail. Most products miss to indicate catalogue numbers.

Response 4: Following the reviewer’s recommendation, a more detailed description of the technical procedures, and the catalogue number of the products used in the study, have been added. A new section titled “Study design” has been included (pages 12 to 16).

Minor considerations

Point 5: Although Fig. 1 is probably shown representative illustration, additional images should be available as supplementary Figures.

Response 5: Following the reviewer’s recommendation, additional images of all experimental conditions are shown now as Supplemental Figure 2.

Point 6: It is not clear if the authors subcloned DNMTs 3´UTRs in commercial vectors or whether commercial products were available as used. The authors need to explain the reasons to support the use of 2 different Luciferase constructs. Also, the positions of the 3´UTRs of the referred genes in their corresponding mRNAs (was the whole sequence or a fragment of it?) and the restriction sites used in the subcloning.

Response 6: We have addressed the reviewer`s query, and more details about the 3’UTR-DNMTs have been now added in the “Cell culture and Luciferase assay” section included in the revised version of the manuscript (page 15, lines 615 to 61834).

Dr. Letizia Pitto´s laboratory kindly donated 3’UTR-DNMT3a plasmid. They validated that miR-30c can downregulate DNMT3a by directly interacting with DNMT3a 3’UTR in an in vitro assay. Therefore, we thought that using a validated plasmid showing the direct interaction between 3’UTR-DNMT3a and miR-30c was the best option for our study. Regarding 3’UTR-DNMT3b, to our knowledge, there is no available publication showing a direct interaction with miR-30c. Therefore, we bought a commercial 3’UTR-DNMT3b pLight plasmid and the empty vector from Switchgear Genomics.

We did not perform subcloning, and both 3’UTRs are fragments of the sequence as follows:

  • 3’UTR-DNMT3b

Restriction pair:  Nhe1/Xho1

Insert length:  1500 bp

  • 3’UTR-DNMT3a

Restriction pair:  SmaI/SacI

Insert length:  1525 bp

Point 7: On line 203 the sentence “DNMT3a and DNMT3b were direct mRNA targes of miR-30c-5p in HeLa cells.” has to be rephrased as there is no evidence for a direct mechanism when the variable measured is luciferase activity.

Response 7: We have now rewritten the sentence (page 6, lines 236 to 237).

Point 8: Explain RE and ROD abbreviations in Fig. 2

Response 8: We have added proper explanations of RE and ROD meaning in the Figure 2 legend (page 6, lines 213 and 218).

Point 9: Please indicate the n for each group in Fig. 2 legend, either 2 or 3 in each case

Response 9: The N has been indicated for each group in the Figure 2 legend (page 6, line 220)

Point 10: Intronic miRNAs are generally referred as mirtrons.

Response 10: Following the reviewer´s recommendation, the term "Intronic miRNAs” have been replaced by mirtrons term throughout the manuscript.

Point 11: Fig. 4 misses to indicate the n for each reading.

Response 11: The N for each group is indicated in the Fig 4 legend (page 8, line 281)

Point 12: Abbreviations should be used consistently after their introduction. Point 2.3 can be shortened by applying this rule.

Response 12: Following the reviewer´s recommendation, we have shortened the 2.3 section by using abbreviations. (pages 4 to 5, lines 165 to 207).

Point 13: Line 155 It is not “the maintenance that methylates” it is the enzyme.

Response 13: The sentence has been corrected to ensure its correct meaning (page 4, line 169).

Point 14: Latin terms such as “de novo” should be italicized. Please review the whole sentence on lines 154 and following.

Response 14: Following the reviewer´s recommendation, the term "de novo” have been italicized throughout the manuscript.

Point 15: Please check typo in “Merk”

Response 15: The typo has been corrected.

Reviewer 2 Report

Dear authors,

It has really impressed me, and it covers the majority of aspects that should be addressed in this study; however, there are a few issues you need to address before I make a final decision.

1- This study needs to provide graphical representations of the whole study.

2- In the material and method section, there should be a section explaining the study design.

3-In fig. 2 There is nothing clear about 2 C, D, G, and H. Try to find other plots that represent your data. Figure 2G is especially unacceptable.

4- Section 4.9 should cover the whole process of statistical analysis, the process of statistical analysis should be outlined in detail, and you should explain how you select your method of analysis (one way, two way, or three way). In that case, you will have to add a section explaining how your study was designed.

5- It seems that I am missing the conclusion part of your paper that concludes your study and explains how you intend to proceed forward. 

Author Response

It has really impressed me, and it covers the majority of aspects that should be addressed in this study; however, there are a few issues you need to address before I make a final decision.

Many thanks for your valuable comments to help us to improve our manuscript.

Point 1: This study needs to provide graphical representations of the whole study.

Response 1: Following the reviewer’s indication, the authors have added a graphical representation of the whole study as Supplemental Figure 4 in the methods section (4.2 study design).

Point 2: In the material and method section, there should be a section explaining the study design.

Response 2: Following the reviewer’s query, a new section named “Study design” has been added to the methods section in the revised version of the manuscript (page 13, lines 492 to 509).

Point 3: In fig. 2 There is nothing clear about 2 C, D, G, and H. Try to find other plots that represent your data. Figure 2G is especially unacceptable.

Response 3: The comments of the reviewer have allowed us to realize that the graphs showing the protein expression quantification on the top of the western blots were not very helpful for the readers. Therefore, we decided to remove the graphs leaving only the blots in the new Figure 2. Furthermore, mean, SEM and n values have been now added to the result section (page 5, lines 193 to 205).

Point 4: Section 4.9 should cover the whole process of statistical analysis, the process of statistical analysis should be outlined in detail, and you should explain how you select your method of analysis (one way, two way, or three way). In that case, you will have to add a section explaining how your study was designed.

Response 4: Following the reviewer´s recommendation, the previous section 4.9, (section 4.10 in the new version of the manuscript), has been now rewritten, and more details about the statistical design of the study have been added (pages 15 to 16, lines 627 to 643).

Point 5: It seems that I am missing the conclusion part of your paper that concludes your study and explains how you intend to proceed forward. 

Response 5: Following the reviewer’s comment, we have added a conclusion of the study and a future perspective (page 12, lines 469 to 477).

Round 2

Reviewer 1 Report

The improvements and additions to the reviewed version of the manuscript are positively valued by this reviewer.

However, additional minor changes are considered necessary to further clarify some parts to readers as follows:

Although the materials & methods section indicates: “The mechanical threshold was defined as the force (g) required to elicit 50% of positive responses.” The force is not measured in g, please correct. Explain what is conceived as a positive response.

No explanation (justification) of the different applied pressures: 0.6; 1; 1.4 …..100 (as displayed on Fig S1 was found. Please include correct force units applied throughout the manuscript.

The supplementary files provided include:

-          1. A word file with some legends that are indicated as Figures (not supplementary) which appear unintelligible and out of order (start with Figure 1B¿?, where is Figure 1A? Figure legends should be self-descriptive (abbreviations included). The significance of labels such as F(1,116) ..etc are not clear in the present form.

-          2. An Excel displaying a long list of tabs named as Figures showing numbers without units or any indication of their meaning.

 It is all quite confusing.

In the new text (lines 125 and following) the authors once more miss to indicate what do shown numbers correspond to. Although methods indicate “mean fluorescence intensity” the reader should not be asked the task of searching on the methods to find out what is being reported. So, the word “mean” should be included for clarity. Please use consistently “mean” instead of “average”, as in Fig 1 legend to prevent confusion.

Please make sure to include details to describe what variable is being measured, corresponding detection method and its units for each experiment.

The new text on lines 240 and following, should replace the control combinations after “compared with cells transfected with the scramble miR-mimic”. In its actual form is not clear what is experimental and what is control, as the controls are included after the text: “of either DNMT3a-3'UTR” “or DNMT3b-3’UTR” but then the control vectors are mentioned in the text between brackets.

The new sections of Fig. 3 need to have different nomenclature (cannot be two Figs 3A, two 3B). Please fix it correspondingly in the text also. A good option would be to move the controls so that they appear side by side with the corresponding experimental data, then with a single Fig 3A (all pGL) and a B (all pLight) labels.

For the new text introduced on lines 79 and following: does cite 14 include the animal protocol described? If not so, please indicate the citation following the detailed method and previous findings.

The addition of a “study design” section in Methods is considered very convenient for the reader to appreciate the overall work in the study. The corresponding illustration (Fig. S4) even helps further. It is noticed, though, that the added text in the Methods section needs extensive English edition. Also, the “N” studied should be indicated in the illustration. A full figure legend should be added.

Author Response

Response to Reviewer 1 Comments

The improvements and additions to the reviewed version of the manuscript are positively valued by this reviewer.

Many thanks for your valuable comments to help us to improve our manuscript.

However, additional minor changes are considered necessary to further clarify some parts to readers as follows:

Point 1: Although the materials & methods section indicates: “The mechanical threshold was defined as the force (g) required to elicit 50% of positive responses.” The force is not measured in g, please correct. Explain what is conceived as a positive response.

Response 1: Following the reviewer’s indication, we have explained what is considered a positive response in the von Frey test, in the methods section (page 13, lines 503 to 514) in the new version of the manuscript. Furthermore, the force now is expressed in millinewtons (mN) throughout the manuscript.

Point 2: No explanation (justification) of the different applied pressures: 0.6; 1; 1.4 …..100 (as displayed on Fig S1 was found. Please include correct force units applied throughout the manuscript.

Response 2: Following the reviewer’s indication, we have added an explanation of the different applied monofilaments in the methods section (page 13, lines 503 to 507). As indicated above, the force now is expressed in mN. Consequently, Figure S1 has been now modified in the new version of the manuscript.

Point 3: The supplementary files provided include:

  1. A word file with some legends that are indicated as Figures (not supplementary) which appear unintelligible and out of order (start with Figure 1B¿?, where is Figure 1A? Figure legends should be self-descriptive (abbreviations included). The significance of labels such as F(1,116) ..etc are not clear in the present form.
  2. An Excel displaying a long list of tabs named as Figures showing numbers without units or any indication of their meaning.

It is all quite confusing.

Response 3: We apologize if the supplementary files provided were confusing. To clarify and facilitate the reader's understanding of the amount of data in our study, we have added a pdf file containing for each figure the raw data together with the statistical analysis. Furthermore, an explanation of the results included has been now added with the appropriate units in each case.

Point 4: In the new text (lines 125 and following) the authors once more miss to indicate what do shown numbers correspond to. Although methods indicate “mean fluorescence intensity” the reader should not be asked the task of searching on the methods to find out what is being reported. So, the word “mean” should be included for clarity. Please use consistently “mean” instead of “average”, as in Fig 1 legend to prevent confusion.

Response 4: Following the reviewer’s recommendation, we have now added the word “mean” where corresponds, and the word “average” has been replaced with mean.

Point 5: Please make sure to include details to describe what variable is being measured, corresponding detection method and its units for each experiment.

Response 5:Folowing the reviewer’s query, the variable, its units, and the detection method have been now added to the new version of the manuscript.

Point 6: The new text on lines 240 and following, should replace the control combinations after “compared with cells transfected with the scramble miR-mimic”. In its actual form is not clear what is experimental and what is control, as the controls are included after the text: “of either DNMT3a-3'UTR” “or DNMT3b-3’UTR” but then the control vectors are mentioned in the text between brackets.

Response 6: Following the reviewer’s recommendation, it has been now modified in the new version of the manuscript (page 6, lines 220 to 228).

Point 7: The new sections of Fig. 3 need to have different nomenclature (cannot be two Figs 3A, two 3B). Please fix it correspondingly in the text also. A good option would be to move the controls so that they appear side by side with the corresponding experimental data, then with a single Fig 3A (all pGL) and a B (all pLight) labels.

Response 7: Following the reviewer’s indication, we have modified the text related to the Luciferase results and figure 3 in the new version of the manuscript. However, the authors consider that the controls and the experimental conditions results should be together in a graph for each DNMT studied.

Point 8: For the new text introduced on lines 79 and following: does cite 14 include the animal protocol described? If not so, please indicate the citation following the detailed method and previous findings.

Response 8: Cite 14 corresponds to our previous study in which the animal protocol is described.

Point 9: The addition of a “study design” section in Methods is considered very convenient for the reader to appreciate the overall work in the study. The corresponding illustration (Fig. S4) even helps further. It is noticed, though, that the added text in the Methods section needs extensive English edition. Also, the “N” studied should be indicated in the illustration. A full figure legend should be added.

Response 9: Following the reviewer’s indication, the N values have been added to Figure S4 together with a legend. The manuscript has been edited by Helena Kruyer, a native English scientist with more than 30 years of experience in proofreading and editing scientific articles for publication in English-language journals. The certificate is included with the supplemental material.
